# The Use of Modern Technologies by Dentists in Poland: Questionnaire among Polish Dentists

**DOI:** 10.3390/healthcare10020225

**Published:** 2022-01-25

**Authors:** Mateusz Świtała, Wojciech Zakrzewski, Zbigniew Rybak, Maria Szymonowicz, Maciej Dobrzyński

**Affiliations:** 1Pre-Clinical Research Centre, Wroclaw Medical University, Bujwida 44, 50-345 Wrocław, Poland; wojciech.zakrzewski@student.umw.edu.pl (W.Z.); zbigniew.rybak@umw.edu.pl (Z.R.); maria.szymonowicz@umw.edu.pl (M.S.); 2Department of Pediatric Dentistry and Preclinical Dentistry, Wroclaw Medical University, Krakowska 26, 50-425 Wrocław, Poland

**Keywords:** dentistry, computer-controlled local anesthetic delivery, ultrasounds, chemomechanical caries removal, modern technologies, laser, ozone

## Abstract

Background: From one year to another, dentists have access to more procedures using modern techniques. Many of them can improve the effectiveness of dental procedures and frequently facilitate and accelerate them. Objectives: Technically advanced devices are an important part of modern dentistry. Over the years, there were developed technologies like ultrasounds, lasers, air abrasion, ozonotherapy, caries diagnostic methods, chemomechanical caries removal (CMCR), pulp vitality tests, computer-controlled local anesthetic delivery (CCLAD). The aim of this study was to investigate the requirement of Polish dentists for such technologies. Methods: An anonymous questionnaire was posted on a social media group of dentists from Poland. 187 responses were obtained. Results: It turned out that almost every respondent uses ultrasounds, but other technologies are not as popular. 43% use CCLAD, 33% use diagnostic methods, 28% use air abrasion, 25% use dental lasers, 21% use CMCR, 18% use pulp vitality tests and 6% use ozonotherapy. The most common reason for not using the aforementioned technologies were their high cost and the sufficient effectiveness of raditional methods. There was a correlation between use of a dental laser and CCLAD and size of office, CMCR use and dentists’ work time and air abrasion use and gender. Many dentists claim that they will try one of the modern technologies in the future. Conclusions: It can be concluded that Polish dentists tend to use ultrasounds and CCLAD more than any other technology. In the future this may change, so more studies in this topic are needed.

## 1. Introduction

Dentistry is a branch of medicine which dynamically develops new technologies. From one year to another, dentists have access to more procedures using modern techniques. Many of them can improve and the effectiveness of dental procedures. And accelerate the related procedures. Some of the technologies were introduced to dentistry years ago. For example, ozone in the form of ozonated water was used in dentistry for the first time by Dr. E.A. Fisch and in surgery by Dr Erwin Payr. They reported their results in 1935 [1]. Another technology that was also developed years ago is lasers, which were introduced in the 1960s by Miaman [2].

On the other hand, some technologies are quite new. Lussi et al., in 1999, validated the use of the DIAGNOdent system (KaVo, Biberach, Germany) for the detection and quantification of caries on occlusal surfaces [3]. The pulse oximetry was invented by Takuo Aoyagi in the early 1970s and in 2007 V. Gopikrishna et al. constructed a pulse oximeter dental probe for assessment of pulp vitality [4]. A computer-controlled local anesthetic delivery system (CCLAD) was introduced in 1997 by Milestone Scientific Inc. as the Wand [5]. 

This shows that dentists need to constantly keep up on their knowledge about new technologies. In times where the Internet is a great source of knowledge, practitioners have a great opportunity to learn about newly developed technologies and adapt them to their work. The aim of this study was to investigate modern technique use by Polish dentists by posting an online survey in a social-media group for Polish dentists. Another purpose of study is to give information, which can be utilized to streamline the usage of technologies in Polish dental practices. The authors tried to identify factors which correlate with new technology use patterns (like sex, years in practice, size of office). The authors assume that there will not be statistically important difference in terms of sex and expect a difference in terms of years in practice and size of office. We also identified reasons which might lead dentistry practitioners to stop using new technologies. 

## 2. Materials and Methods

An anonymous questionnaire was posted in a social media group of Polish dentists in January 2019. Before publication, the aspects of privacy and data security were addressed. The survey was planned to be anonymous. The survey only concerned opinions and the expression of thoughts of respondents and not clinical trials on humans. Due to this, it was considered unnecessary to proceed with the formal approval procedures. The generation of the questionnaire and collection of responses were done through Google Forms (Google LLC, Mountain View, CA, USA). Members were given four weeks to respond. 

The survey had 10 sections. The first section included questions about the biometric data of the respondents. In this section, the authors asked about gender, years in practice, type of employment and size of their offices. Respondents with more than one year in practice were considered in this study. Sections 2 to 9 had five questions each, investigating the use of each technology, including laser, ultrasounds, air abrasion, ozone, diagnostic methods, chemo-mechanical caries removal, pulp vitality tests, and computer-controlled local anesthetic delivery. The tenth section included general questions about reasons for not using the mentioned technologies, patients attitudes about new technologies, and sources of knowledge about new technologies (Table 1). The online form is visible in [6].

Collected data were tabulated using Microsoft Excel (Microsoft, Redmond, WA, USA). Only the authors had access to data collected from the survey. 

All analyses were performed with the help of Statistica for Windows (version 13.3, TIBCO Software, Palo Alto, CA, USA). A chi-squared test was used to assess the statistical significance. Probabilities less than 0.05 were accepted as statistically significant.

## 3. Results

187 responses were acquired over January and February 2019. The biometrics of the respondents are shown in Figure 1. A significant portion of the respondents were women (82.9%), which reflects gender proportion of dentists in Poland. There were roughly equal responses in terms of years in practice groups, with fewer dentists in the >20 years in practice group. Most of the respondents worked in an office with two to five units. A major portion of the respondents (79.5%) had their own office. 82.9% of respondents did not have any specialization. There were 10 dentists with prosthetic specialization, 10 dentists with restorative specialization, seven with surgery, three with orthodontics, two with pedodontics, and two with periodontics. 

The percentage use of each investigated technology is shown in Table 2. 

The ratings of effectiveness and difficulty of each technology are shown in Figure 2, Figure 3, Figure 4, Figure 5, Figure 6, Figure 7, Figure 8 and Figure 9. Most dentists rate ultrasounds as less difficult (59%) and more effective (76%) in comparison with manual methods. Air abrasion was rated by the most dentists as comparatively difficult in comparison with the traditional method. In case of other methods, the most respondents didn’t know how difficult and effective they are. 

Procedures in which dentists use ultrasounds are shown in Figure 3b. The most popular ultrasonic procedure is scaling (96%). Endodontic treatment (canals irrigation) (85%) and prosthetic procedures (teeth preparation for crowns, inlays etc., post-and-cores removal) (80%) are popular fields to use ultrasounds. This technology is used by fewer respondents in surgery (piezosurgery) (24%) and in caries removal (32%).

The most respondents use lasers in surgery (18%) and teeth whitening. The least popular laser procedure is caries removal (4%) (Figure 2b).

Respondents use ozone for the following procedures: surgery (wounds disinfection, dry socket, abscesses)–5.36%, periodontology (gingivitis, periodontitis)–3.57%, endodontic treatment (canals disinfection)–3.57%, prosthetic (disinfection of prepared teeth)–1.79% (Figure 5b).

The most popular caries diagnostic method (CDM) is FOTI/Di-FOTI (27%) (Figure 6a). 

In our survey, we also asked dentists if they would use each technology in the future. Almost every respondent claimed that they may use ultrasounds (99%). Also, many dentists show interest in using other technologies in the future: CDM (91%), laser (86%), pulp vitality tests (PVT) (86%), air abrasion (85%), CCLAD (82%), ozone (80%), chemo-mechanical caries removal (CMCR) (64%). 

A dental laser was present in the offices of only 30% of respondents. All dentists had an ultrasonic device in their offices.

An air abrasion unit was present in 35% of offices. Only 7% of dentists has access to a device for ozonotherapy.

85.7% of respondents didn’t know about one to five of the methods mentioned in the questionnaire, 9% didn’t know about six to ten of the techniques, and six percent didn’t know about more than ten of them. There wasn’t a statically relevant difference between groups of dentists with different work schedules.

In our study we also investigated the reasons for not using modern technologies in dental practices. The most dentists (51%) don’t use a dental laser because of the high cost. High cost was also reason for not using ozone (41%), caries diagnostic methods (30%), pulp vitality tests (30%) and computer-controlled local anesthetic delivery (29%). Lack of knowledge was a reason for not using a dental laser for 9% of respondents, ozonoterapy for 12%, caries diagnostic methods for 10%, and chemo-mechanical caries removal for 14%. 

The sufficient effectiveness of traditional methods was a reason to not use air abrasion (22%), caries diagnostic methods (CDM) (21%), CMCR (28%), PVT (29%) and CCLAD (24%).

Because of higher complexity in use in comparison with traditional methods, respondents do not use CDM (20%), PVT (16%), CMCR (14%), air abrasion (11%) and ozonoterapy (8%).

For respondents, the most popular source of knowledge were: courses (89%), the Internet (75%), science magazines and research (69%), knowledge earned during college, specialized courses (42%), and books (41%).

A majority of respondents (57%) say that their patients show interest in innovative methods of dental treatment. 39% claim that method of treatment is inert for their patients and only 4.5% respondents claim that their patients refuse treatment with these methods.

A chi-squared test showed statistical differences in the use of CMCR in different years in practice groups. Dentists that have worked for more than 20 years seems to not use CMCR. Differences in laser use in groups of office sizes was also statistically important. In smaller offices with only one dental unit, lasers were less popular than in bigger offices. The same situation was observed with regard to CCLAD use patterns. A statistical assessment showed that female dentists use air abrasion more often than male dentists–Table 3.

## 4. Discussion

In 2012, Verma, S. et al. predicted that specific laser procedures would become essential components of contemporary dental practice over the next decade [7]. In 2019, the dental laser was present in only 30% of Polish offices and 75% dentists did not use the laser in their practice. Mentioned authors in research review showed that laser can be used in many dental procedures, for example: cavity preparation, caries and restorative removal, treatment of dentinal hypersensitivity, and surgical procedures. Cozean et al. concluded that using the Er:YAG laser is both safe and effective for caries removal and cavity preparation [8]. Valenti et al. evaluated the ability of the Er:YAG laser in reducing the microbial population in carious lesions. The authors showed that the use of lasers resulted in greater reduction of bacterial CFU than traditional preparations [9]. These conclusions don’t affect the use of lasers by Polish dentists for the mentioned procedures, as only 3.75% of respondents use it. 

Verma, S. et al. mentioned surgical procedures which can be performed with the use of a laser, which include aesthetic gingival re-contouring, crown lengthening, exposure of unerupted teeth, removal of inflamed, hypertrophic tissue, and frenectomies [7]. Laser dental surgery was used by 17.85% dentists.

The authors claim that these procedures, with the help of a laser, can be performed without bleeding, sutures and with no need for special postoperative care. This correlates with the opinion of Polish dentists: 40.2% of them think that laser procedures are less or as difficult as classical ones and 37.4% dentists claim that this procedure has better or equal effectiveness.

Schwarz et al. investigated the ability of lasers to treat dentinal hypersensitivity. He proved that the Er:YAG laser is effective in this procedure, and that the results lasted longer than when the traditional procedure was used [10]. For this procedure, 14.28% of respondents use a laser. In conclusion, the research reveals that a laser can be a valuable method in modern dentistry and can enhance the effectiveness of many procedures, but it is not a popular method among Polish dentists.

In a review of the literature, Walmsley et al. pointed out that research showed that both methods, manual and ultrasonic, are effective in calculus removal on a clinical level, but scanning microscopy studies have suggested that ultrasonic methods are more effective. Ultrasonic scalers are also more effective in the more inaccessible areas of the oral cavity such as the posterior molars [11]. Polish dentists agree with that opinion, as 95.54% of them use ultrasounds for scaling and 75.9% rate it as more effective. 

In the opinion of Plotino et al., in endodontic treatment, ultrasounds can be used for access refinement, finding calcified canals, removal of intracanal obstructions, activation of irrigating solutions, and root canal preparation. The authors mention many benefits of ultrasonic procedures, such as improved visualization combined with a more conservative approach when selectively removing tooth structure, specific angulation or tip design [12]. In 2020, Abu Hasna A et al. showed that passive ultrasonic irrigation decreased levels of *Enterococcus faecalis, Esherichia coli* and endotoxins in combination with NaOCl [13]. Verma et al. provided a randomized controlled trail of endodontic treatment with the help of ultrasonic and laser-activated irrigation. They observed a 100% success rate of periapical periodontitis healing after endodontic treatment with the addition of additional activation [14]. Thanks to their effectiveness, ultrasonic procedures are very popular in Polish dentistry. 84.82% of respondents use it for endodontic treatment.

Another field of dentistry where ultrasounds can be used is surgery. Thomas M. et al. mention oral and maxillofacial surgery procedures, which can be done with the help of piezosurgery. These are: sinus lift, bone graft harvesting, periodontal surgery, cyst removal, ridge expansion, osteogenic distraction, unilateral condylar hyperplasia, dental extraction and impacted tooth removal [15]. Agarwal et al. show advantages of piezosurgery as being precise and selective bone cutting, faster healing, less invasive, reduced post-operative pain, and better tactile sensitivity [16]. As the largest disadvantage, the authors consider increased operating time [15,16]. These observations were also confirmed by Otake Y et al. In an experimental study, the authors determined a difference in the time of osteotomy done with piezosurgery and rotary instruments. Piezosurgery required three times longer to cut the bone, but did not cut soft tissues [17]. In Poland, piezosurgery is not as widely used. Only 24.11% of respondents use it. 

Hegde VS and Khatavkar RA mention indications for air abrasion: removal of superficial enamel defects, detection of pit and fissure caries suspect, preparation of cavities restricted only to small section of the tooth, surface preparation of abfractions and abrasions, the removal of existing restorations, and the avoidance of local anesthesia [18]. Air abrasion is used by only 28% of questioned Polish dentists. In our study, 60.9% of dentists rate air abrasion as effective as traditional techniques. Similar conclusions were drawn by Bhushan and Goswami. In their study, air abrasion pretreatment did not result in a statistically significant difference in sealant retention in both primary and permanent molars after three and six months follow-up [19].

Ozonotherapy was an area of research for William C. Domb. He gathered indications for using ozone in dentistry. These were: treating caries, periodontal disease, endodontics treatment, perioral viral and fungal infections, sinusitis and even temporomandibular joint dysfunctions. He also takes ozone as proper treatment in osteonecrotic lesions after bisphosphonate medications [20]. Gupta and Mansi had reviewed many cases where ozone was successfully used in periodontal disease. They confirm the disinfecting ability of ozone and suggest using it in daily dental practice [21]. In Polish dentistry, ozone seems not to be popular, as only 6.25% of respondents use it, (3.57% in periodontology). This may change, as 80% claim that they will or may use ozone in the future.

Gomez J. discusses the current available methods to detect early caries lesions. She mention methods like quantitative light-inducted fluorescence (QLF), DIAGNOdent, fibre-optic transillumination (FOTI) and its digital version–Di-FOTI, and electrical conductance measurement (ECM). In the opinion of the authors, these methods should be used as an adjunct to well-established and evidence-based methods such as visual assessment and radiographs [22]. In our study, 32.14% respondents use one of the caries diagnostic methods. Cho KH et al. showed that quantitative light-induced fluorescence (QLF) has the ability to detect caries of occlusal surfaces in primary teeth [23]. In the opinion of 52% of our respondents, new caries diagnostic methods are more or comparatively effective in comparison with visual or radiographic methods. 

Chemomechanical caries removal (CMCR) seems to be the optimal method in treating caries, especially in children who are anxious about dental procedures. In Venkataraghavan and Karthik et al.’s study, the use of CMCR resulted in decreasing pain complaints and the reduced need for anesthesia. The only disadvantage of CMCR was an increase in cleaning duration [24]. Similar conclusions was drawn by Sontakke, Priyanka et al. in 2019. The authors report an overall absence of bad smell/taste in CMCR [25]. A CMCR was compared to Er:YAG and carbid burs in terms of the ability to remove microorgams from cavities. CMCR showed the lowest ability and the Er:YAG laser was the most capable of decreasing the size of the cavital biome [26]. In a systemic review, Cardoso et al. compared efficacy and patient acceptance of caries removal with alternative methods. Traditional preparation showed faster caries removal and resulted in larger cavities, which can lead to the unnecessary removal of healthy tissues. Rotary instrumentation often was related with a need for anesthesia. Patients were experiencing less negative emotions (pain, fear) when alternative methods were used [27]. 78.6% of our respondents do not use chemomechanical methods in their dental practice. Efficiency of CMCR was rated as “less effective” by 24.5% of questioned dentists and 56.6% of them did not know how effective this method is. The low popularity of CMCR may be a result of the lack of knowledge or experience about it among respondents. 

Pulp vitality testing is an important step in endodontic treatment. It determinates the best option for treatment. In 2017, Salgar, AR et al. rated electrical tests in comparison with thermal tests. He has shown that thermal tests are a better option in pulp vitality diagnosis than electrical tests [28]. Mainkar A. et al., compared five dental pulp tests: cold pulp testing (CPT), heat pulp testing (HPT), electric pulp testing (EPT), laser Doppler flowmetry (LDF), and pulse oximetry (PO). In their study, LDF and PO were the most accurate diagnostic methods and should be used by clinicians if possible. HPT was the least accurate diagnostic method [29]. In Poland, most of our respondents use traditional cold or heat tests. Only 17.86% use one of the newer tests (EPT, LDF or PO). From these, EPT is used predominately. 26% of surveyed dentists say that these tests are more reliable than the traditional cold test. 

Aggarwal, K et al. compared anxiety and pain levels during local anesthesia using traditional syringe and computer-controlled local anesthetic delivery (CCLAD). In their study, patients reported lower anxiety levels during CCLAD anesthesia. 64.4% of patients preferred CCLAD [30]. Mittal M. et al. have shown, in their study, that intraligamentary anesthesia can be more effective and less painful for children with the help of CCLADS devices [31]. Flisfisch S et al. evaluated patients opinions after local anesthesia with CCLAD and a conventional syringe. Most of the patients rated CCLAD as more acceptable [32]. Pozos-Guillén et al. delivered a meta-analysis about children’s pain and fear levels during dental local anesthesia with the use of a standard syringe and CCLAD. The analysis shows that lower levels of negative emotions occurred when CCLAD was used [33]. In our study, 42.86% of respondents use CCLAD, but only 10.8% of them think that it is more effective than local anesthesia with the conventional syringe. Most of Polish dentists (73%) say CCLADS systems are as effective as the syringe.

One of the limitations to the study is that almost 65% of respondents fell into the category of having <5 or 5–10 years of practice, so they could have founding limitations. This can be the reason why more costly technologies are not used as frequently. Practitioners with more years in practice do not use social media as often as their younger colleagues, which could explain the low number of respondents in the category of >20 years of practice. The low numbers of obtained responses is a major limitation in this study. Further studies need to include other options of survey dissemination. 

### Publications Which Compare Use of Other Modern Technologies by Dentists in Europe

Nassar HM et al. conducted a survey about novel caries diagnostic technologies among restorative dentists. In their study, most dentists chose optical translumination (FOTI/DIFOTI) as the preferred method, saying that it has the widest clinical usage (i.e., for detecting enamel cracks) and is easy to use. The main reason for rejecting other methods was their high cost [34]. These findings greatly complement the results of our survey.

D. Tran et al. formed 1031 online surveys that were sent to a sample of UK dentists. 385 practitioners responded. Most users did not use any CAD/CAM technology and the main barriers to use this technology were, according to them, the lack of perceived benefit and initial costs as disadvantages. CAD/CAM technology was mainly used by dentists delivering private work. Most users of CAD/CAM technology were trained either by themselves or by companies, but on the other hand, a significant number of CAD/CAM users felt that their training was insufficient. 89% of respondents think that CAD/CAM has an important role in the future of dentistry [35]. 

Van der Zande et al. investigated the degree of digital technology use among general dental practitioners. A questionnaire was created that has reached 1000 practitioners in the Netherlands. The response rate was 31.3%. Dentists have adopted an average number of 6.3 ± 2.3 technologies. 22.5% were low technology users (0–4 technologies), 46.2% were intermediate technology users (5–7 technologies) and 31.3 were high technology users (8–12 technologies). What was interesting was that high technology users were younger on average (*p* = 0.024), had invested more hours per year in professional activities (*p* = 0.026), were more likely to have a specialization (*p* < 0.001), and also worked for more hours per week (*p* = 0.003) than low technology users. Among technologies that were asked about in a questionnaire were digital intraoral radiography, digital orthopantomogram, digital 3D radiography CBCT, intraoral camera and scanner, CAD/CAM systems, and others. According to the questionnaire, out of the nonclinical technologies, digital registration of patient information is the most frequently used technology (93.2%).

When it comes to clinical and diagnostic technologies, digital intraoral radiography (90%) and digital orthopantomograms (57.2%) are used most often. 

The authors of the questionnaire confirm the importance of such a study, saying “Understanding where dentistry is going in terms of digital developments begins with knowing where dentistry stands now, and how digital technologies are incorporated at present.” [36].

## 5. Conclusions

In Poland, dentists tend to use ultrasounds the most. Other technologies are not as popular. This may change in the future, as many dentists say that they will try some of the new technologies. In most cases, the usage of each technology did not depended on the size of office, work experience or sex. The most common reason for not using modern technologies was their high cost. This study showed that even if reports say some technologies are a better option, dentists prefer using conventional ones.

From the above data, it follows that the use of new technologies reduces dental procedure duration and makes treatment more effective. They allow for the detection of diseases in earlier stages, which directly relate to the reduction of therapy costs for patients and for the health care system (insurance system).

The authors are convinced that this research is an important addition in understanding the current state of technologies in dentistry.

## Figures and Tables

**Figure 1 healthcare-10-00225-f001:**
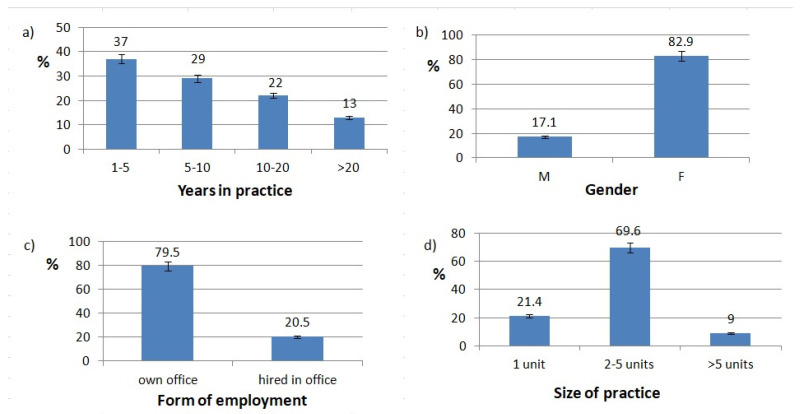
Respondents biometrics. (**a**) Years In practice distribution, (**b**) gender of responding dentists, (**c**) dentist’s form of employment, (**d**) size of practice.

**Figure 2 healthcare-10-00225-f002:**
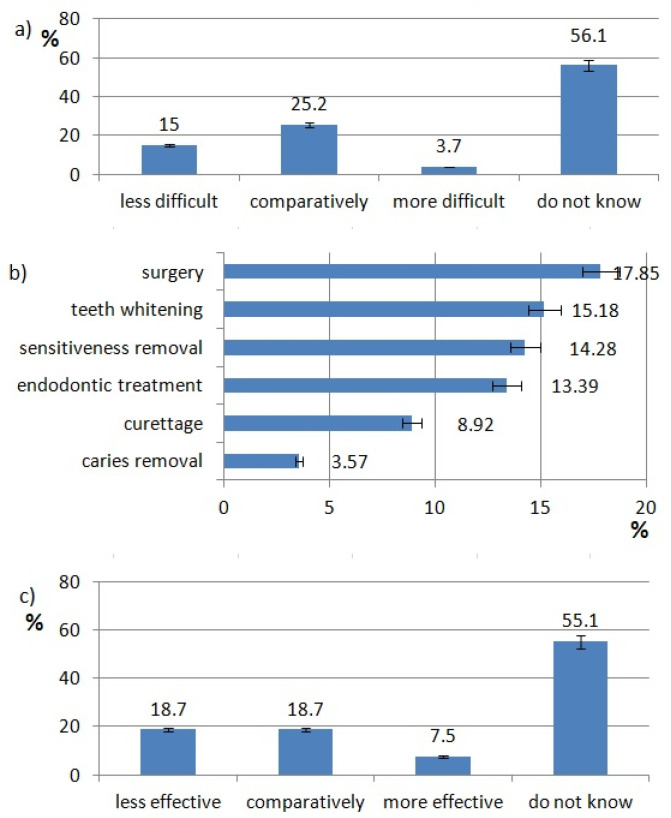
Laser use pattern of responding dentists. (**a**) difficulty of laser procedures, (**b**) type of laser procedures, (**c**) effectiveness of laser procedures.

**Figure 3 healthcare-10-00225-f003:**
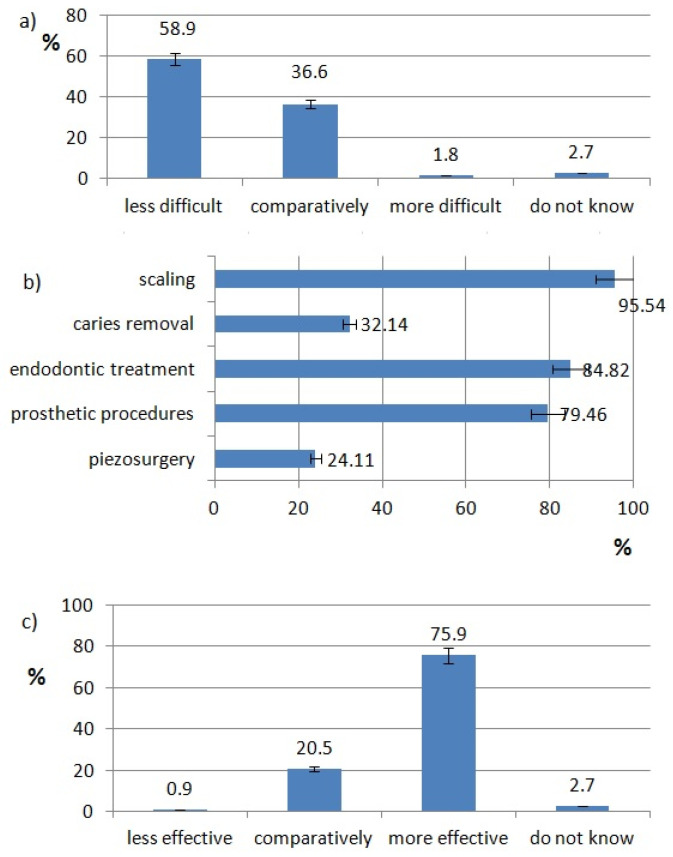
Ultrasounds use pattern of responding dentists; (**a**) difficulty of ultrasonic procedures, (**b**) type of ultrasonic procedures, (**c**) effectiveness of ultrasonic procedures.

**Figure 4 healthcare-10-00225-f004:**
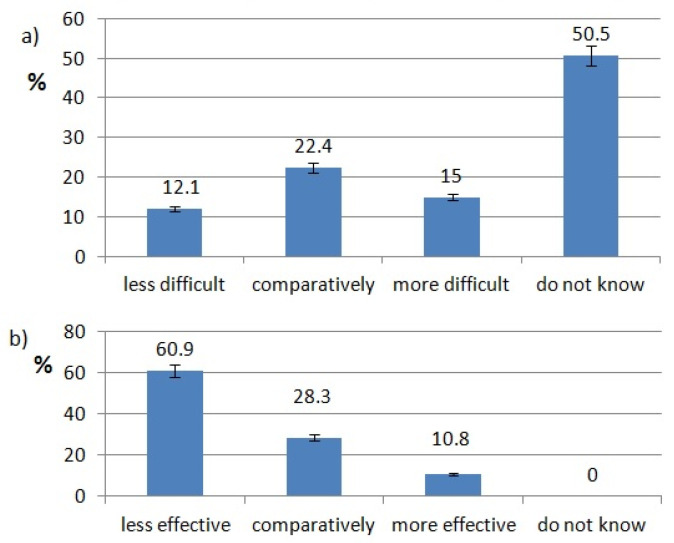
Air abrasion use pattern of responding dentists; (**a**) difficulty of air abrasion procedures, (**b**) effectiveness of air abrasion procedures.

**Figure 5 healthcare-10-00225-f005:**
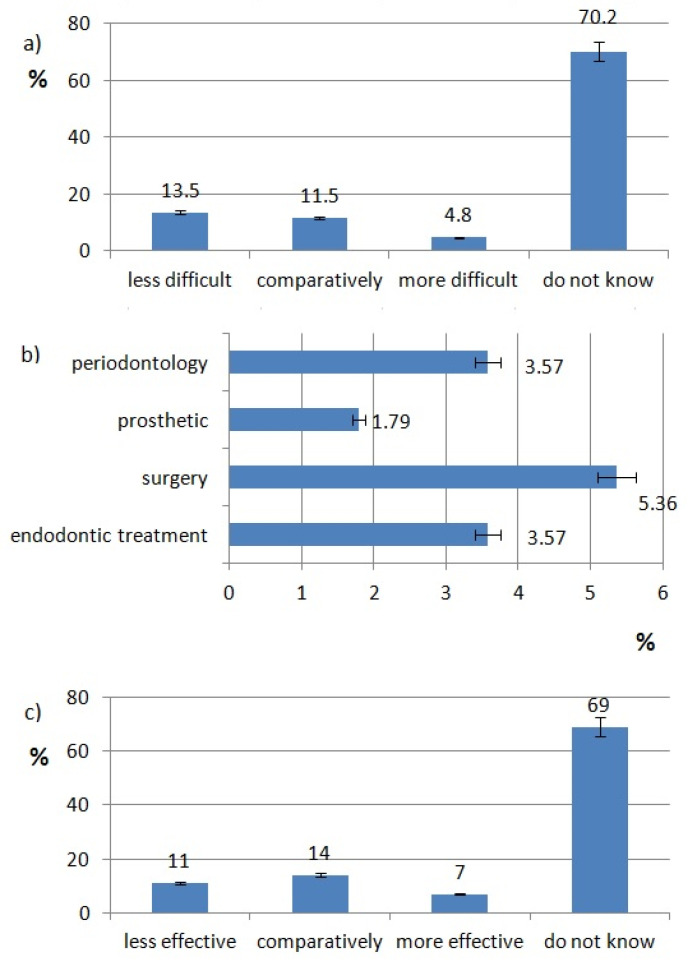
Ozone use pattern of responding dentists; (**a**) effectiveness of ozone procedures, (**b**) type of ozone procedures, (**c**) difficulty of ozone procedures.

**Figure 6 healthcare-10-00225-f006:**
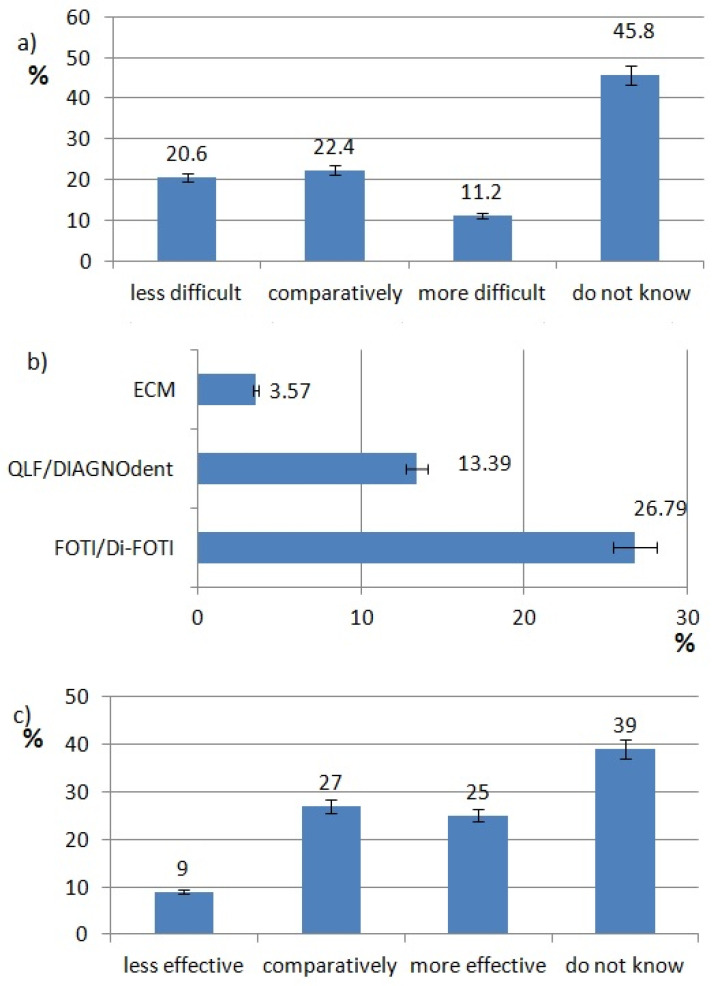
Caries diagnostics methods use pat tern of responding dentists; (**a**) use of each method, (**b**) difficulty of caries diagnostics methods, (**c**) effectiveness of caries diagnostics methods.

**Figure 7 healthcare-10-00225-f007:**
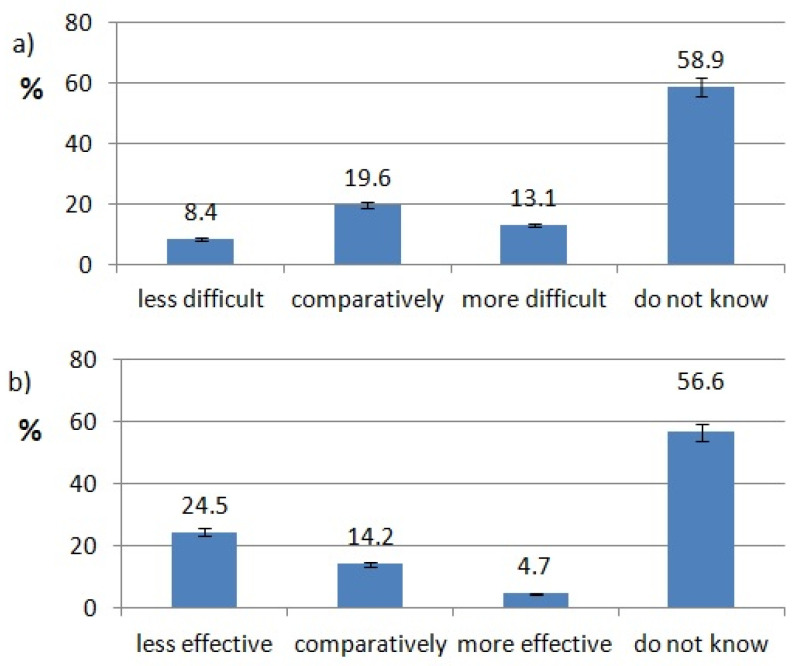
Chemo-mechanical caries removal use pattern of responding dentists; (**a**) effectiveness of CMCR, (**b**) difficulty of CMCR.

**Figure 8 healthcare-10-00225-f008:**
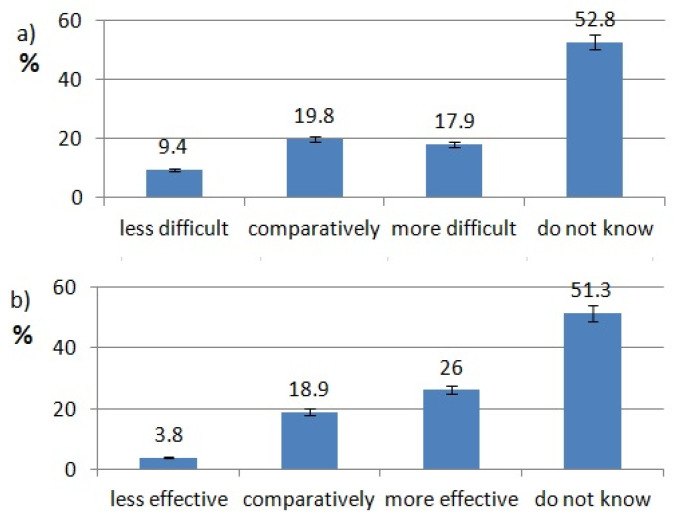
Pulp vitality tests use pattern of responding dentists; (**a**) effectiveness of pulp vitality tests, (**b**) difficulty of pulp vitality tests.

**Figure 9 healthcare-10-00225-f009:**
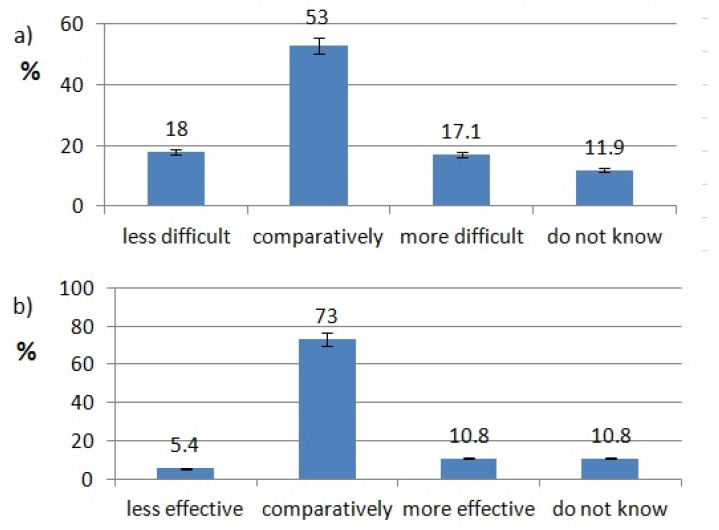
Computer controlled local anesthetic delivery use pattern of responding dentists; (**a**) effectiveness of CCLAD, (**b**) difficulty of CCLAD.

**Table 1 healthcare-10-00225-t001:** Questions included in the survey.

Questions in Survey	Responds
Biometric questions	
1. Gender (choice question)	Male/Female
2. Years in practice (graded question)	<5/5–10/10–20/>20
3. specialization (multiple choice question)	- restorative dentistry - dental surgery - periodontology - orthodontics - pedodontics - prosthetics - no specialization
4. Form of employment (choice question)	- own office - hired in office
5. Size of office (graded question)	one unit/two to five units/>five units
Questions in each technology section	
1. Have you access to this technology in your practice? (graded question)	Yes/No
2. How often do you use this technology? (graded question)	- don’t use - less often than conventional method - as often as conventional method - more often than conventional method - use only modern method
3. How do you rate efficiency of this technology? (graded question)	- More efficient - as efficient as conventional method - less efficient - don’t know
4. How do you rate difficulty of this technology? (graded question)	- More difficult - as difficult as conventional method - less difficult - don’t know
5. Will you use this technology in the future? (choice question)	Yes/No/Maybe
General questions	
1. About how many methods you didn’t know? (graded question)	1–5/6–10/>10
2. Why don’t you use mentioned technologies? (multiple choice question)	- high cost - more complicated method - less efficient method - sufficient effectiveness of conventional method - lack of knowledge
3. What is the attitude of patients to innovative technologies? (choice question)	- They are interested - They are inert - They refuse treatment with these technologies
4. Which sources of knowledge do you use to acquire competency about mentioned technologies? (multiple choice question)	- journals/research - courses - college/ specialization courses - internet - books - other (open question)

**Table 2 healthcare-10-00225-t002:** Percentage of respondents who declare use of each technology.

Technology	Percentage of Use
Laser	25
Ultrasounds	97
Air abrasion	28
Ozone	6
CDM	33
CMCR	21
PVT	18
CCLAD	43

**Table 3 healthcare-10-00225-t003:** The number of respondents (percentage) who use each technology.

CCLAD	PVT	CMCR	CDM	Ozone	Air Abrasion	Ultrasounds	Laser	Technology	
33 (43)	11 (16)	20 (29)	18 (26)	2 (3)	20 (29)	65 (94)	13 (19)	<5 (*n* = 69)	Years in practice
23 (43)	7 (13)	8 (15)	18 (33)	6 (11)	15 (28)	54 (100)	17 (31)	5–10 (*n* = 54)	
11 (27)	8 (20)	12 (29)	15 (37)	2 (5)	13 (32)	41 (100)	12 (29)	10–20 (*n* = 41)	
13 (57)	7 (30)	0 (0)	11 (48)	2 (9)	5 (22)	22 (96)	12 (29)	>20 (*n* = 23)	
0.08	0.03	0.009 *	0.261	0.288	0.862	0.142	0.373	P	
8 (20)	7 (18)	7 (18)	15 (38)	2 (5)	8 (20)	37 (93)	2 (5)	1 unit (*n* = 40)	Size of office
64 (49)	23 (18)	28 (22)	42 (32)	8 (6)	41 (32)	128 (98)	41 (32)	2–5 u. (*n* = 130)	
8 (47)	3 (18)	5 (29)	5 (29)	2 (12)	3 (18)	17 (100)	4 (24)	>5 u. (*n* = 17)	
0.004 *	0.999	0.603	0.782	0.619	0.224	0.096	0.003 *	P	
67 (43)	28 (18)	33 (21)	55 (35)	12 (8)	50 (32)	152 (97)	37 (24)	F (*n* = 157)	Gender
13 (43)	5 (17)	7 (23)	7 (23)	0 (0)	3 (10)	30 (100)	10 (33)	M (*n* = 30)	
0.947	0.878	0.777	0.212	0.118	0.015 *	0.322	0.259	P	

CDM–caries diagnostic methods, CMCR–chemo-mechanical caries removal, PVT–pulp vitality tests, CCLAD–computer-controlled local anesthetic delivery. P–based on chi squared test, * *p* < 0.05.

## Data Availability

Not applicable.

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
