# Peer review of "The Use of Modern Technologies by Dentists in Poland: Questionnaire among Polish Dentists"

_healthcare, 2022, doi:10.3390/healthcare10020225_

Round 1

Reviewer 1 Report

Switala et al, presents a review of modern dental technologies preferred by Polish dental practitioners, the data which has been gathered from posting questionnaire on the social media. The study is informative and can be utilized to streamline usage of technologies in Polish dental practices. One of the major limitations of this study is low number of respondents and almost >65% of respondents falling into the category of either <5 years or 5-10 years of practice, who might have funding limitations to set-up more expensive technologies and relies on more conventional methods of treatments such as ‘ultrasound’. A good indicator for the cost limitation is a correlation between the years of practice, Units and usage of modern/ more expensive technologies. In addition, the authors may address this in their discussion section. Nevertheless, the current information gathered may be of importance in fine-tuning current dental diagnostic/ treatment technologies for future usage.

Line 81: It would be informative for the readers what was the minimum years of practice/ units per year that was considered for this study. As a practitioner with less or equal to 1 year of practice might not be very informative.

Overall, the article is well-written, however there are few typing errors that needs to be corrected such as:

Line 35: ‘Ozon’ check spelling

Line 55: ‘V’

Line 79: Please correct all decimal points from ‘comma’ to ‘dot’

Author Response

Dear Reviewer,

We would like to express our sincerest gratitude to the Reviewers for their enormous efforts in criticizing the manuscript. All remarks have been included in the revised version of the manuscript.

Question 1

Switala et al, presents a review of modern dental technologies preferred by Polish dental practitioners, the data which has been gathered from posting questionnaire on the social media. The study is informative and can be utilized to streamline usage of technologies in Polish dental practices. One of the major limitations of this study is low number of respondents and almost >65% of respondents falling into the category of either <5 years or 5-10 years of practice, who might have funding limitations to set-up more expensive technologies and relies on more conventional methods of treatments such as ‘ultrasound’. A good indicator for the cost limitation is a correlation between the years of practice, Units and usage of modern/ more expensive technologies. In addition, the authors may address this in their discussion section. Nevertheless, the current information gathered may be of importance in fine-tuning current dental diagnostic/ treatment technologies for future usage.

Answer: We would like you for the comment. Information about limitation mentioned in this question was addressed in discussion.

Question 2

Line 81: It would be informative for the readers what was the minimum years of practice/ units per year that was considered for this study. As a practitioner with less or equal to 1 year of practice might not be very informative.

Answer: We would like to thank you for the comment. The minimum years of practice considered for this study was 1+ year. Information about this was included In “Methods and Materials” section.

Question 3               

Line 35: ‘Ozon’ check spelling

Answer: We would like to thank you for the comment. The word has been corrected.

Question 4

Line 55: ‘V’

Answer: We would like to thank you for the comment. ‘V’ in line 55 is a shortcut of first name of cited author.

Question 5

Line 79: Please correct all decimal points from ‘comma’ to ‘dot’

Answer: We would like to thank you for the comment. All decimal points was corrected from ‘comma’ to ‘dot’.

Reviewer 2 Report

I think that this article must be radically changed. Its topics fits the Healthcare policy.

In my opinion, the aim of the article must be clearer presented. On the other hand, "Discussion" must be reconsider. It is not necessary to describe each technique but to adapt them to the targeted polish population. However, this discussions must be adapted to the results , that need to be more refined. 

Of course, the "conclusions" will be changed up to he new aspect of the article. 

Author Response

Dear Reviewer,

We would like to express our sincerest gratitude to the Reviewers for their enormous efforts in criticizing the manuscript. All remarks have been included in the revised version of the manuscript.

Question 1

In my opinion, the aim of the article must be clearer presented. On the other hand, "Discussion" must be reconsider. It is not necessary to describe each technique but to adapt them to the targeted polish population. However, this discussions must be adapted to the results , that need to be more refined. Of course, the "conclusions" will be changed up to the new aspect of the article. 

Answer: We would like to thank you for the comment. The aim of article have been expanded and results refined. In ‘Discussion’ we wanted to confront the effectiveness of each method and the opinion of polish practitioners about them. Despite our willingness, we could not reach other articles targeting polish population in viable databases. ,Discussion’ section have been also refined. Changes have been made in ‘Conclusions’ section.

Reviewer 3 Report

I don't think the introduction is adequate. It seems like a long list of materials, with no connection whatsoever. There is no reference to the null hypothesis of the study.

Who took the quiz? Was it previously validated?

The results are very extensive. There are graphics that have little information and/or that are repeated in the text. Graphs should have absolute values. There has to be a better characterization and stratification of the sample, namely by age. There will always be differences regarding the use of new technologies at different ages.

The discussion should refer to the limitations of the study.

The conclusion is too long. The suggestions given at the end should be discussed.

Author Response

Dear Reviewer,

We would like to express our sincerest gratitude to the Reviewers for their enormous efforts in criticizing the manuscript. All remarks have been included in the revised version of the manuscript.

Question 1

I don't think the introduction is adequate. It seems like a long list of materials, with no connection whatsoever. There is no reference to the null hypothesis of the study.

Answer: We would like to thank you for the comment. Changes have been made in ‘Introduction’ section.

Question 2

Who took the quiz? Was it previously validated?

Answer: We would like to thank you for the comment. The quiz was taken by first author. All authors had their part in creation of survey and validation.

Question 3

The results are very extensive. There are graphics that have little information and/or that are repeated in the text. Graphs should have absolute values. There has to be a better characterization and stratification of the sample, namely by age. There will always be differences regarding the use of new technologies at different ages.

Answer: We would like to thank you for the comment. Results have been refined. Graphics which had information doubled in the text have been removed.

Question 4

The discussion should refer to the limitations of the study.

Answer: We would like to thank you for the comment.  We added information about limitations of the study in ‘Discussion’.

Question 5

The conclusion is too long. The suggestions given at the end should be discussed.

Answer: We would like to thank you for the comment. The conclusion was shortened.

Reviewer 4 Report

 Technically advanced devices are important part of modern dentistry. Over the years, there were developed technologies like ultrasounds, lasers, air abrasion, ozonotherapy, caries diagnostic methods, chemomechanical caries removal (CMCR), pulp vitality  tests, computer-controlled local anesthetic delivery (CCLAD). The aim of the study proposed by the authors  was to investigate, the requirement of polish dentists for such technologies.  The authors developed an electronic  anonymous questionnaire that was posted on social-media group of dentists from Poland. They obtained 187 responds. Results showed that almost every respondent uses ultrasounds, but other technologies are not as much  popular. 43% use CCLAD, 33% caries diagnostic methods, 28% air abrasion, 25% dental lasers, 21%  CMCR, 18% pulp vitality tests and 6% ozonotherapy. The most common reason for not using mentioned technologies were their high cost and sufficient effectiveness of traditional method. The authors concluded  that polish dentists tend to use ultra-sounds and CCLAD more than any other technology. In the future this may change, so more studies  in this topic are needed.

The article is interesting.

This reviewer suggests the following improvements:

1) Introduction

"The aim of this study was to investigate modern techniques use by polish dentists."

I suggest expanding the "purpose" and inserting a short presentation of the study structure.

2) More details are needed in relation to privacy, regulations, cyber security and design features.

See for example the methods in https://www.mdpi.com/2227-9032/9/10/1347 (You do not need to add the reference)

4) It is necessary to detail step by step the type of question used (open question, choice, multiple choice, graded, likert….)

5) An example of the survey with the link and QR code could be useful both to the reader and to reviewers to understand the significance of the study.

6) Please details the output of the Chi squared test (for example of the 43% use CCLAD). You can add this for example  in the Table 2.

7) All the figures need improvement. For example they need data labels and error bars.

Author Response

Dear Reviewer,

We would like to express our sincerest gratitude to the Reviewers for their enormous efforts in criticizing the manuscript. All remarks have been included in the revised version of the manuscript.

Question 1

"The aim of this study was to investigate modern techniques use by polish dentists."

I suggest expanding the "purpose" and inserting a short presentation of the study structure.

Answer: We would like to thank you for the comment.  Your suggestion was included in ,,Introduction “ section.

Question 2

More details are needed in relation to privacy, regulations, cyber security and design features.

See for example the methods in https://www.mdpi.com/2227-9032/9/10/1347 (You do not need to add the reference).

Answer: We would like to thank you for the comment.  Your suggestion was addressed in ‘Materials an Methods’ section.

Question 3

It is necessary to detail step by step the type of question used (open question, choice, multiple choice, graded, likert….)

Answer: We would like to thank you for the comment. Your suggestion was addressed in Table 1., where we detail questions and viable answers included in our survey.

Question 4

An example of the survey with the link and QR code could be useful both to the reader and to reviewers to understand the significance of the study.

Answer: We would like to thank you for the comment.  An reference to the link to the survey was included.

Question 5

Please details the output of the Chi squared test (for example of the 43% use CCLAD). You can add this for example  in the Table 2.

Answer: We would like to thank you for the comment.  The Chi squared test was used to determine if there is statistical difference in usage of each technology in groups of practitioners with different years in practice/gender/size of office. This is presented in Table 3.

Question 6

All the figures need improvement. For example they need data labels and error bars.

Answer: We would like to thank you for the comment.  The figures have been improved with data labels and error bars. Figures with little information added to text or information doubled in text have been removed.

Round 2

Reviewer 2 Report

I think that changes made into the text are suitable to transform the present article in a version that can be published

Reviewer 3 Report

Thank you for your effort in answering all of the reviewers' concerns.

Reviewer 4 Report

The authors have improved the ms according to the suggestions